# Potato Planter and Planting Technology: A Review of Recent Developments

Baidong Zhou, Yexin Li, Cong Zhang, Liewang Cao, Chengsong Li, Shouyong Xie and Qi Niu *

College of Engineering and Technology, Southwest University, Chongqing 400715, China
* Correspondence: niuqi2019@swu.edu.cn; Tel.: +86-152-0107-1052

**Abstract:** Potato is one of the most important food crops in the world, which is of great significance for sustainable agricultural development. Mechanized planting is the essential technical link in mechanized production, which has an essential component in the potato growing industry. The mechanization of potato planting technology is an effective method of increasing potato yields. A variety of potato planting technologies and machines have been developed around the world. This review presents the research progress and application status of potato planters and planting technology worldwide. It classifies the planting technology into four types: research of materials characteristics for potatoes, soil cultivation, seed potato separation, and zero-speed seeding. The most critical seed potato separation technology was divided into six types according to the structure of the seed metering device. Detailed features have been provided for some typical potato planters and soil cultivation machines. Finally, the developing trend of intelligent planting technology was analyzed, and some suggestions were proposed to promote the development of potato planters.

**Keywords:** potato planter; soil cultivation; mechanization; seed metering device; intelligent seeding

## 1. Introduction

Potato (*Solanum tuberosum* L.), originated from the Andes of Peru in South America and was introduced into Europe through Spain, Asia, and other parts of the world through Europe [1]. At present, potatoes are grown and produced in more than 160 countries worldwide, the fourth-largest food in the world after wheat, rice, and corn [2].

Potato planting areas in China are widely distributed in 29 provinces. Differences in natural conditions, agronomic requirements, and cultivation conditions have led to significant differences in the degree of mechanization [3]. The largest potato-producing areas in the United States are distributed in the three northwest States, and the second-largest concentrated producing regions are distributed in the four northern states [4]. Relying on appropriate planting temperature, fertile soil, modern processing equipment, and professional experience inherited from generation to generation, the American potato industry has always been in a leading position worldwide. In Europe, the British potato granules distributed in the temperate and oceanic climate are advantageous to the potato from extreme temperatures, plenty of rainfall, and nutrient-rich soil ensure potato growth. Dutch dominance in kinds of potato production and production system to its highest level of potato production in the world. Potato production areas in Russia are mainly distributed in the northwest, which is a black soil area with fertile soil, which is conducive to potato growth [5]. Potato production in Japan is at a high level. It is in a leading position in Asia, with a yield per unit area similar to that in Europe and the United States. The main production areas are concentrated in Hokkaido, which belongs to a one-season potato cultivation area [6].

In recent years, the global potato planting area has declined, but the output is still increasing. With the continuous decline of the sown area and production in Europe, the United States and the continuous development in Asia and other regions, the global potato

production center has gradually shifted from developed to developing countries [7]. The major potato producers in the world are the USA, China, Germany, Russia, etc. Some countries' planting situation and yield data in recent years are shown in Tables 1 and 2.

**Table 1.** Harvested Area /production of potatoes in different countries. Date from reference [8]. Copyright 2022 FAO.

| Year | China | U.S.A. | Britain | Germany | Russia | Category |
|------|-------|--------|---------|---------|--------|----------|
| 2015 | 4788.1 | 426.7 | 129.0 | 236.7 | 2111.6 | |
| 2016 | 4805.1 | 419.9 | 139.0 | 242.6 | 1425.6 | |
| 2017 | 4862.4 | 422.7 | 146.0 | 250.5 | 1335.6 | Harvested Area/ |
| 2018 | 4760.7 | 414.1 | 140.0 | 252.2 | 1313.5 | thousand ha |
| 2019 | 4038.9 | 379.3 | 144.0 | 271.6 | 1238.6 | |
| 2020 | 4218.2 | 369.9 | 142.0 | 273.5 | 1178.1 | |
| 2015 | 82,893.2 | 20,012.7 | 5644.3 | 10,370.3 | 33,645.8 | |
| 2016 | 84,986.5 | 20,426.4 | 5394.7 | 10,772.2 | 22,463.5 | |
| 2017 | 88,536.4 | 20,453.4 | 6218.0 | 11,720.4 | 21,707.6 | Production/ |
| 2018 | 90,321.4 | 20,607.3 | 5027.7 | 8921.0 | 22,395.0 | thousand tons |
| 2019 | 75,657.9 | 19,251.3 | 5307.0 | 10,602.2 | 22,074.9 | |
| 2020 | 78,236.6 | 18,789.9 | 5520.0 | 11,715.1 | 19,607.4 | |

**Table 2.** Yield of potatoes in different countries. Date from reference [8]. Copyright 2022 FAO.

| Year | China | U.S.A. | Britain | Germany | Russia | Category |
|------|-------|--------|---------|---------|--------|----------|
| 2015 | 17.31 | 46.90 | 43.75 | 43.81 | 15.93 | |
| 2016 | 17.69 | 48.65 | 38.81 | 44.40 | 15.76 | |
| 2017 | 18.21 | 48.39 | 42.59 | 46.79 | 16.25 | Yield/t/ha |
| 2018 | 18.97 | 49.76 | 35.91 | 35.37 | 17.05 | |
| 2019 | 18.73 | 50.75 | 36.85 | 39.04 | 17.82 | |
| 2020 | 18.55 | 50.79 | 38.87 | 42.83 | 16.64 | |

Potato planting is the most important component of potato production. However, the cultivation modes and natural conditions are various due to the difference in the production of potatoes. In the complicated agronomic process, labor intensity also affects production efficiency [9]. Therefore, mechanized planting is crucial for improving production efficiency and reducing labor [10,11]. In plain areas, it is suitable for the operation of large and medium-sized sowing machinery [12] (as shown in Figure 1a), providing convenient conditions for the development of mechanization. Small potato planters have the characteristics of light weight, convenience, and simple structure, as shown in Figure 1b. These planters are mainly designed for hilly mountainous areas. However, this planter is rarely used nowadays because of its low working efficiency and needs manual assistance [13].

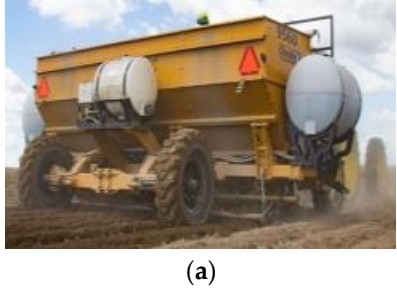
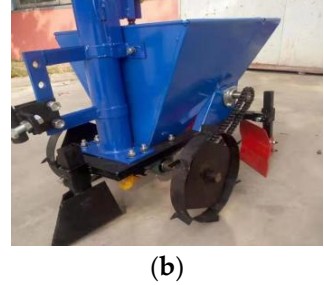

(**a**)         (**b**)

**Figure 1.** Different types of potato planters, they should be listed as: (**a**) large combined potato planter. Reprinted with permission from ref. [12]. Copyright 2021 Double L Industries; (**b**) small potato planter.

This paper introduces the potato planter and planting technology related content: "The key technology status of the mechanized planting" explores the existing key technology and categorized the applications into four types. "Machine of mechanized planting" elaborates on some typical commercial potato planters and soil cultivation machines. "Research of intelligent seeding technology" proposes a systematic analysis of future directions based on intelligent seeding technology. Finally, "Conclusions and recommendations" concludes the mechanized planting along with some current problems and makes some suggestions.

## 2. The Key Technology Status of the Mechanized Planting

In the process of mechanized potato planting, it is necessary to tillage machinery for soil preparation and then use the potato planter for mechanized seeding. According to the agronomic requirements, the potato planter needs one operation to complete the work requirements of ditching, spraying, seeding, fertilizing, and ridging. As the core part of potato planter [14], the performance of the seeding metering device will directly determine the operation quality and working effect of the planter [15,16]. The seeds in the seed hopper are generally arranged in disorder. It is necessary to pick up the potato seeds from the seed hopper through the seed metering device, separate them, and put them into the seed bed. The technical core of mechanized seeding is the orderly single granulation and the maintenance of a single granulation state, that is, the technology of seed tuber separation and whole row picking and stable seed guide [17]. In the process of potato plant design, it is often necessary to analyze the material characteristics of seed potatoes [18,19]. Therefore, according to the elements of the working process of mechanized planting, the critical technologies of mechanized planting can be divided into the research of material characteristics for potatoes, soil cultivation, the seed potato separation, and the seed guiding technology of the planter.

### 2.1. Research of the Material Characteristics of Seed Potatoes

The characteristics of potatoes are directly related to the design of critical components of the seed metering device, which affects the potato production level [20]. To design and optimize the device structure, the physical characteristics have been researched according to different types of potatoes. Generally, the physical attributes of seed potato can be divided into two aspects: essential physical characteristics and dynamic characteristics.

2.1.1. Division of Seed Potato types

In addition to differences in the planting areas, there are also differences in the types of seed tubers selected, which can be divided into the whole tuber, sliced tuber, and mini-tuber [21]. Figure 2 presents three kinds of seed tubers. At present, the most commonly used potato in China and India is the sliced potato, which has the advantages of being low cost and easy to obtain, but it needs to be graded, whole row cutting, spraying and lubricants, and other processes need to be carried, and it is easy to cause potato cross-infection in the treatment which will directly affect the yield. Whole potato seeding can avoid cross-infection caused by cutting, and causes minimal damage to seed potato and high germination rate, which is suitable for mechanized operation, but requires special cultivation [22]. Mini-tuber is a kind of potato produced by virus-free seeding, with a neat shape, uniform size, and excellent quality. It can be directly used for seeding, and the seedling emergence rate can reach 100% [23]. The main reason for the low yield of potato production in some regions is the small application area and low penetration rate of virus-free mini-tubers. In contrast, high-quality virus-free min-tubers are widely used in high-yield areas of developed countries.

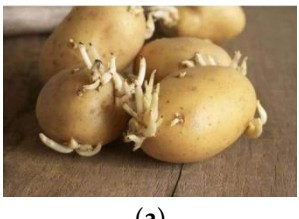
(**a**)

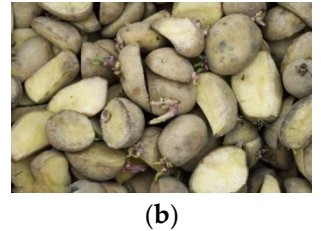
(**b**)

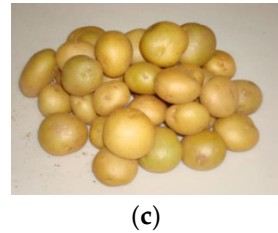
(**c**)

**Figure 2.** Types of potato: (**a**) whole tuber; (**b**) sliced tuber; (**c**) mini-tuber.

2.1.2. Research on the Basic Physical Characteristics of Seed Potatoes

The basic physical characteristics of potatoes mainly include triaxial size, shape, and density. It is an essential basis for designing the structure parameters of seed arrangement device and seed box.

In determining the essential physical characteristics of potatoes, vernier calipers are generally used to measure the overall dimension of potatoes, and the shape index ($f$) of seed tubers is obtained through the formula. Then the shape and size parameters of seeds are defined according to the shape index ($f$). Generally, the key structural parameters of the planter can be determined based on the shape and size parameters [24]. Seed potato moisture content is a crucial characteristic index of seed tuber and has a specific influence on the properties of other materials [25]. Seed potatoes with higher moisture content are easily damaged during the working process. The adhesion can easily affect seed picking performance—the general use of atmospheric constant temperature drying method or moisture meter measurement. To facilitate the theoretical modeling and virtual simulation of critical components, the volume density and monomer density of potatoes should be measured. When the seed potato moisture content is determined, volumetric density can be measured by mass volume ratio, and monomer density can be measured by suspension under liquid immersion [26].

2.1.3. Research of Dynamic Characteristics of Seed Potatoes

Potatoes can be classified as granular materials according to the classification of agricultural materials. The knowledge of the dynamic characteristic of granular material such as friction characteristics, compression characteristics, and flow characteristics between particle–particle or particle–surrounding can help improve the design and operation of the machine.

Before the simulation analysis, the physical model of the seed potato needs to be established. For the material with a tremendous difference in the shape of seed potato, the computerized tomographic (CT) technology [21], and image processing technology [27] were not suitable. DEM (Digital Elevation Model) is one of the advanced numerical methods used to study the dynamic characteristics of granular material, which is widely used in the study of the movement of agricultural granular materials. Since DEM discrete element simulation is based on the theoretic contact mechanical model, the simulation result is inconsistent with the actual one when the software is simulating the essential flow characteristics of the bulk materials. The pile angle can reflect the flow characteristics of the material [28,29]. Liu et al. used the method of combining test and simulation to calibrate the material parameters and established the test simulation model [30]. Seeds could be divided into class of spherical, small ellipsoid, and large ellipsoid according to shapes. The seed shape is shown in Figure 3, and the model is shown in Figure 4. The contracting parameters of the discrete particle were taken as the independent variables, the measurement results of the simulation model were taken as the evaluation, and then the relevant fitting equations were established and the corresponding results in the simulation model were obtained by changing the independent variables. Finally, the simulated contact parameters of the model were obtained by substituting the measured parameters in the actual test into the fitting equation. Due to the irregular shape of potato seeds, its dynamic characteristics in the seed metering device cannot be simulated by selecting a single spherical particle. The models of

large ellipsoidal particles, small ellipsoidal particles, and spherical-like mini-tuber seeds were established by generating particles from a template. The related motion process of the mini-tuber seed adopted a Hertz–Mindlin non-sliding contact mechanical model [30,31]. There are few relevant studies on sliced tuber. Whole potatoes were separated into two longitudinal pieces and the geometric model was established in Solidworks based on the mapping data and imported into EDEM [32].

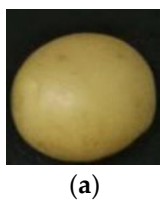 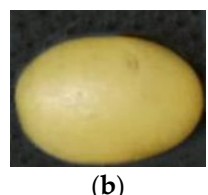 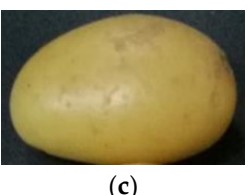

(**a**)         (**b**)         (**c**)

**Figure 3.** Classification of mini-tuber seeds: (**a**) class of spherical; (**b**) small ellipsoid; (**c**) large ellipsoid. Reprinted with permission from ref. [30]. Copyright 2018 China Agricultural University.

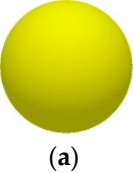 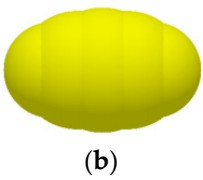 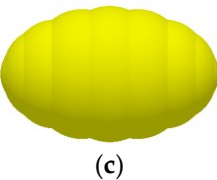

(**a**)         (**b**)         (**c**)

**Figure 4.** Discrete element model of mini-tuber seed: (**a**) class of spherical; (**b**) small ellipsoid; (**c**) large ellipsoid. Reprinted with permission from ref. [30]. Copyright 2018 China Agricultural University.

### 2.2. Soil Cultivation

In the process of mechanized potato planting, the quality of the seedbed has a direct impact on soil water storage capacity, adaptive, and potato yield. The high-quality seedbeds can effectively store water and conserve soil moisture, improve soil permeability, create suitable conditions for early sprout and seedling growth, and facilitate the standardization of field management later.

Potato ridging planting is an effective high-yield planting method. It is also one of the crucial steps of soil cultivation, Figure 5 shows potato ridging planting. Soil cultivation can provide appropriate conditions for ridging planting and improve the seeding quality [33].

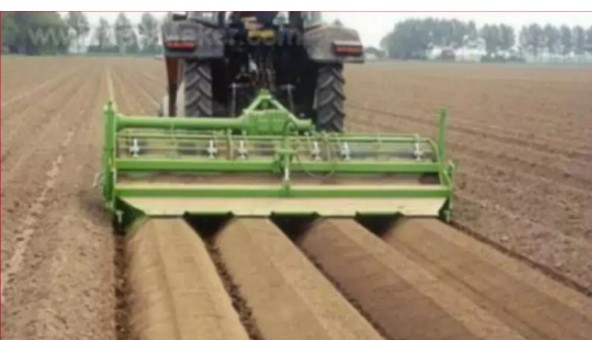

**Figure 5.** Potato ridging planting.

In terms of the selection of the seedbed, a sufficient area convenient for the mechanization of the plot and for the standardization of the potato planting should be chosen. Different soil cultivation practices are usually used in different countries due to conditions. Sub-soiling and power harrow combined tillage is generally adopted in Europe. In contrast, multi-function large-combined tillage planters are mainly adopted in the USA. In addition, rotary tillage and subsoiling machines are the main methods used in Asia.

In the process of mechanized potato planting, a layer of plow bottom will be formed under the tillage layer because the seedbed soil has been compacted by the mechanical equipment plow for a long time. After the previous crop is harvested, the first treatment is sub-soiling, which can break the hard plow bottom. The depth of sub-soiling should be consistent, and the trench bottom should be flat. The coefficient of variation of sub-soiling depth should not exceed 15% [34]. It is usually carried out once every two to four years. Zhao et al. [35] conducted a comparative test on 300 mm sub-soiling, 250 mm sub-soiling, and 200 mm rotary tillage. It was found that the nitrogen, potassium, phosphorus, and other elements in the middle and lower layers of 300 mm sub-soiling were significantly higher than those of 200 mm traditional rotary tillage and 250 mm sub-soiling, which was conducive to the accumulation of nutrients in the lower layer. Similarly, the water content in the lower layer of 300 mm sub-soiling was also better than that of rotary tillage.

### 2.3. Potato Separation

Seed potato separation is one of the critical technologies of mechanized potato planting. It mainly refers to when seed potatoes stacked disorderly in the seed box are formed into a single seed potato through the function of the seed metering device. The technology of seed metering has been actively developed over the past several decades. In terms of the technology status of the seed metering device, the devices can be categorized into six types: cup-belt (chain) type, pneumatic-type, moving-belt-type, needle-type, and pickup-finger-type. Some typical seed metering devices are shown in Table 3.

**Table 3.** Some typical seed metering devices and their principles.

| Type | Principle |
| --- | --- |
| Cup-belt (chain)-type seed metering device | The power of the seed belt (chain) is provided by the ground wheel or hydraulic motor, which drives the driving wheel to rotate through the transmission system. The seeds flow to the feeding area at the bottom of the seed box under gravity. In the forward motion of the seed belt/chain, the spoon mounted on the belt (chain) scoops up the seed potatoes in proper sequence. After the seeds shift to the seed clearing area, the excess materials are removed by the seed clearing device and returned to the seed box. When the cup reaches the highest point, the potato seeds fall on the back of the next cup. Potato seeds are carried by the cups to the seeding point, then dropped into the bottom of the seed furrow [36,37]. |
| Pneumatic-type seed metering device | The seed-suction arm is driven to rotate by power. When the seed-suction arm is connected with the vacuum chamber, the seed-suction arm absorbs a single seed potato from the seed hopper by negative pressure, and stably carries the seed potato to rotate synchronously with the seed metering device. As the potatoes reach the release point, they fall into the seedbed by positive pressure and gravity [17,38,39]. |
| Moving-belt-type seed metering device | When the seeds fall from the seed box to the conveyor belt, the transport speed of the two sides belt is opposite to that of the middle belt, where the middle belt is used to sort the seeds in a single row for seeding, and the two side belts are used to transport and collect the excess seeds to the seed collection port for preparation for subsequent seeding. |
| Needle-stabbed-type seed metering device | The needles are mounted on the circumference of the seeding rotary disk. Each needle stabs and obtains one seed in turns in the seed hopper. With the rotation of the seeding disk to the seed dropping area, the seed potatoes are separated from the needles under the function of scraping components and discharged into seed bed, completing a seeding process [40,41]. |
| Pickup-finger seed metering device | The Pickup-fingers are mounted on the circumference of the seeding rotary disk. Under the elastic force, The pickup-finger is in a normally closed state. The finger arm touches the guide rail and opens in sequence when traveling to a feeding area, and each finger on the vertical disk picks up a seed and moves with it to the next area under rotation condition, the arm meets another set of guide rails, and opens in sequence again [42]. |
| Vibration separation-type seed metering device | The conveyor belt is combined with the principle of forced vibration, and the potatoes are filled into the groove with manual assistance to achieve the function of separation and sorting. The grooves move with the movement of the conveyor belt, and at the release point the potatoes will fall into the seedbed to finish seeding. |



Potato ridging planting is an effective high-yield planting method. It is also one of the crucial steps of soil cultivation. Soil cultivation can provide appropriate conditions for ridging planting and improve the seeding quality [33].

### 2.3.1. Cup-Belt (Chain)-Type Seed Metering Device

The cup-belt(chain) type is the most widely used seed arrangement method worldwide [19]. Compared with the cup-chain type, the cup-belt type (Figures 6–8) is more likely to slip in the process of movement, resulting in the heterogeneous spacing of seeds. Many researchers [37,43] have studied the dynamics characteristics of cup-belt potato planter and reformed its structure. In recent years, several attempts have been made to research the metering process through constructing mathematical model analysis. For example, to define the causation of the deviations in uniformity of placement of the potatoes, a theoretical model was established by Buitenwerf H. et al. [44]. Cai et al. [32] proposed a conical-shaped seed box to abate the missing rate caused by the arching problem for seed potatoes in the conventional potato planter. In order to minimize the reseeding rate and miss-seeding of potato planters, Wang et al. [45] designed a new type of cup-belt type device, which was composed of a motor vibration cleaning system conveyor belt with spoons, etc. Lü et al. [46] designed a cup-belt type device for potato planters and the structure of many vital components. Wollman A. E. et al. [47] added a missing seeding detection system and a reseeding mechanism to the potato planter.

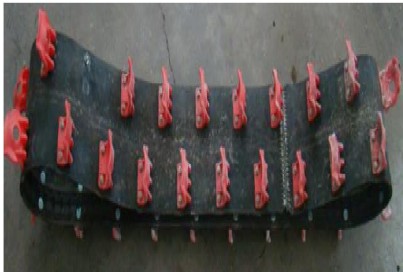

**Figure 6.** Seed metering belt. Reprinted with permission from ref. [48]. Copyright 2020 Chinese Heilongjiang Agricultural Commission.

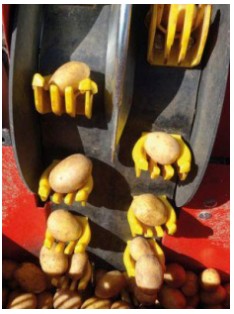

**Figure 7.** Cup-belt type. Reprinted with permission from ref. [49]. Copyright 2019 Miedema Industries.

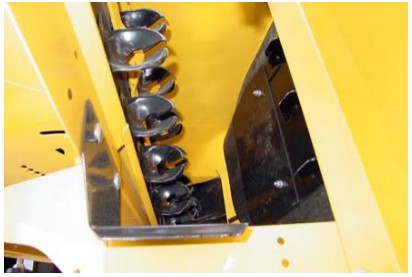

**Figure 8.** Cup-chain type. Reprinted with permission from ref. [50]. Copyright 2021 Double L Industries.

### 2.3.2. Pneumatic-Type Seed Metering Device

The performance of a pneumatic seed metering device is greatly influenced by seed size, shape, the number of holes in the disk, and working pressure. There are several preponderances of these devices, such as low rate of seed damage, high seeding quality, and high planting speed. The negative pressure of seed suction will directly affect the seeding efficiency. Therefore, there are many technical difficulties in potato seeding. Currently, it is mainly used for sowing crops with low weight, such as maize [51], soybean, rapeseed [52,53], etc. According to the different ways of pneumatic seed separation, it can be divided into air suction type, air blowing type, air pressure type, and central gathering type [54]. Compared with other crops, potato seeds have the characteristics of large particle size and irregular shape, and the existing devices are limited to air suction type. Lü et al. [17,18] proposed a potato air-suction metering device (Figure 9) for the north of China.

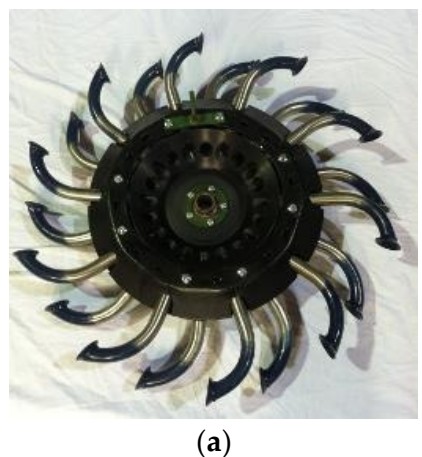 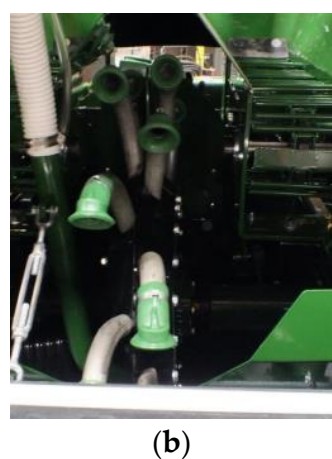

|     |     |
| :-: | :-: |
| (**a**) | (**b**) |

**Figure 9.** Pneumatic-type seed metering device. (**a**) The structure of a pneumatic-type seed metering device. Reprinted with permission from ref. [49]. Copyright 2019 Lockwood Industries. (**b**) Pneumatic-type seed metering device in Lockwood Air Cup Planter. Reprinted with permission from ref. [49]. Copyright 2019 Lockwood Industries.

McLeod C. D. et al. [10] researched and developed a pneumatic micro potato precision seeding device, which can absorb seeds under negative pressure, carry seeds, and row seeds under positive pressure, and a spray gun is set on the seeding device to remove excess seeds and reduce re-seeding. Yang, D. [51] designed a pneumatic horizontal disc potato seed metering device, which uses an intermittent feeding mechanism composed of a grooved wheel mechanism and a transmission chain to realize periodic quantitative seed delivery. Mao et al. [55] designed and optimized a tilting disk pneumatic precision seed metering device for virus-free mini-tuber as shown in Figure 10a. Compared with the traditional mechanical device, the pneumatic seed metering device reduces the mechanical damage and improves the quality of the seed potatoes. Due to the limitation of the working principle and configuration, the device will only miss seeding and will not re-seed. To solve the problem of limited seeding speed for the cup-belt-type seeding device and significant negative pressure required for the pneumatic seed metering device, a mechanical–pneumatic combined metering device for potato (Figure 10b) was designed [49]. The structural parameters and working regulation of the devices were investigated. Based on the theoretical analysis of the potato seed stress, the configuration and parameters of the critical components of the improved seed scoop (Figure 10c) were determined.

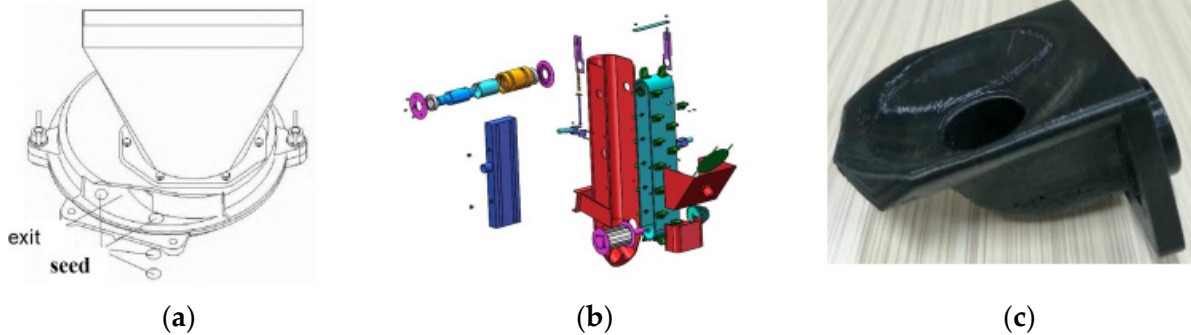

Figure 10. Improved pneumatic-type seed metering device. (**a**) Mao et al.'s research. Reprinted with permission from ref. [55]. Copyright 2013 Huazhong Agricultural University; (**b**) Liu et al.'s research. Reprinted with permission from ref. [49].; (**c**) Improved seed scoop. Reprinted with permission from ref. [49] Copyright 2019 Shandong Agricultural University.

### 2.3.3. Moving-Belt-Type Seed Metering Device

This device (Figure 11) has the advantages of simple structure and principle, small size, low rate of seed damage, and additional vital adaptability to diverse shapes of potato seeds. To reduce the damage rate of potato seeds in the process of mechanized planting, Meijer et al. [56] organized a moving-belt-type seed metering device and conducted experimental research on seeding performance. He et al. [57] created a new type of moving-belt-type belt mechanism and analyzed the overall structure and performance of each factor.

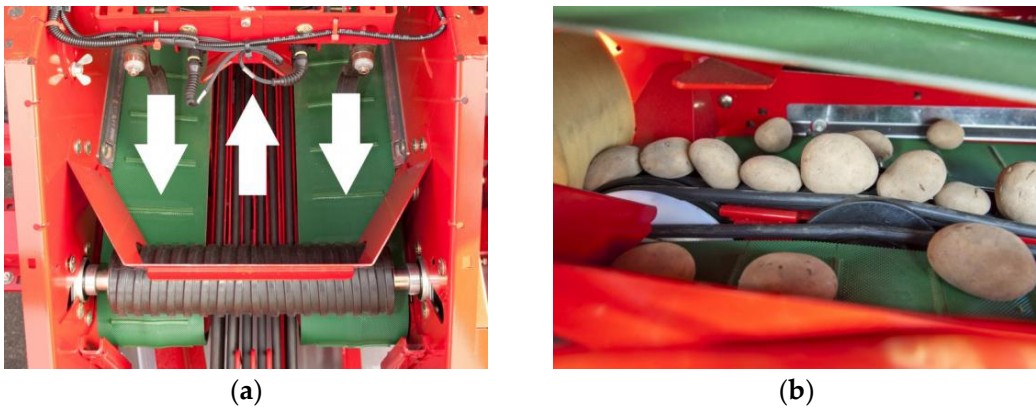

Figure 11. Moving-belt-type seed metering device. (**a**) The conveyor belt arrangement of the device. Reprinted with permission from ref. [58] Copyright 2022 Grimme Industries. (**b**) The working process of the device. Reprinted with permission from ref. [58] Copyright 2022 Grimme Industries.

### 2.3.4. Needle-Type Seed Metering Device

The needle-type uses the needle to pick seed tuber and throw seed, as shown in Figure 12. Although this device has good adaptability to the size and shape of seed tubers, it is easy to cause cross-infection due to the injury of seeds by needle-punched seed metering device and the emergence of virus-carrying seed tubers will cause infection. On the other hand, the impurities mixed in the soil, such as gravel and weeds, can deform the needle and cause damage [59]. Misener G.C. et al. [60] measured the seed piece distribution patterns completed by the cup and needle-type metering devices through numerous experiments. The consequences demonstrated that the needle metering device performed slightly more effectively than the cup type.

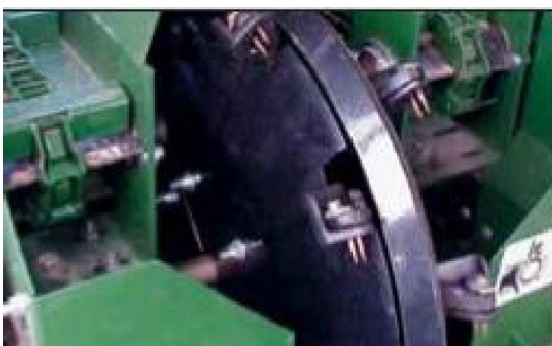

**Figure 12.** Needle-type seed metering device. Reprinted with permission from ref. [51] Copyright 2016 Lockwood Industries.

### 2.3.5. Pickup-Finger Seed Metering Device

This device is suitable for seeding whole or sliced tubers. The pickup-ginger type (Figure 13) is greatly affected by the shape, size, and the seed extraction rate of the seed potato is different at different clamping positions, which requires high classification accuracy for the seed potato. Chen et al. [61] designed a novel device and carried out kinematics and dynamics analysis. Boydas M.G. et al. [62] explored the mechanism of influencing the seeding accuracy of the Pickup-finger device.

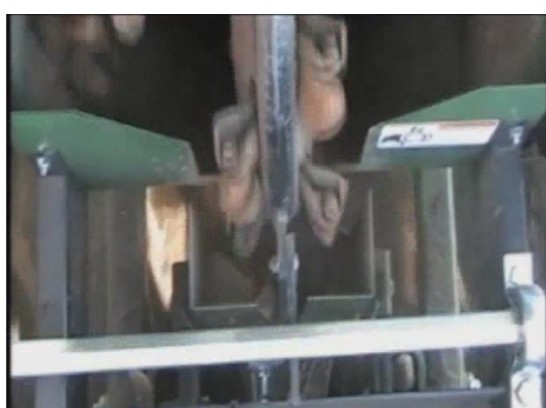

**Figure 13.** Pickup-finger type metering device. Reprinted with permission from ref. [48] Copyright 2016 Lockwood Industries.

### 2.3.6. Vibration Separation Seed Metering Device

Presently, the research on vibration seed metering devices at home and abroad mainly focuses on sowing crops with small particle sizes and regular shapes. Liu et al. [63] invented a planter for mini-tuber established on the principle of forced vibration, as shown in Figure 14. In this device, seeds were arranged and moved to the conveyor belt under the function of vibration plate, and single sequence of seeds was achieved under the restriction of device structure. Excess potatoes would be transported to the conveyor belt for sorting again under the action of vibration plate. The pressure belt was designed in the seed dropping channel for positioning, and the seed was sent to the seed dropping point under the cooperative effort of the conveyor belt and the pressure belt for final seeding.

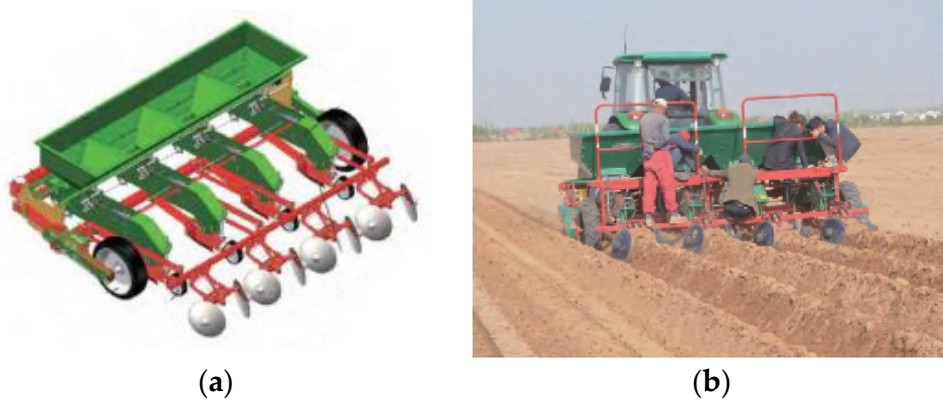

(a)                                    (b)

**Figure 14.** Vibration-arranging-based planter. (**a**) The structure of vibration-arranging-based planter. Reprinted with permission from ref. [63]. Copyright 2019 China Agricultural University. (**b**) The working process of the vibration-arranging-based planter. Reprinted with permission from ref. [63]. Copyright 2019 China Agricultural University.

### *2.4. Zero-Speed Seeding*

The seed guiding step is a connecting link between the preceding step of seed discharging from the seed metering device and the following step of dropping in the seedbed, which affects the orderly state of seeds in the planting process and determines the uniform distribution of subsequent seeds in the seedbed [64]. The seed potatoes will be offset or even jump on the ground, which will reduce the uniformity of plant spacing and sowing accuracy when the planting position is too high, or the horizontal velocity of seed potatoes is relatively high. Due to the large and irregular shape of seed potatoes, this negative effect is more serious. Therefore, reducing the height of seeding and the speed of seed tuber relative to the seedbed along the horizontal direction are essential ways to ensure the quality of seeding. Based on the type and technical features of seed metering device, zero-speed seeding technology of planter can be divided into low position seeding, seed guide tube seeding, and air blowing seeding.

### 2.4.1. Low Position Seeding

Low position seeding means that seed potatoes are transported to a lower position and dropped to the seedbed by carrying seed components through a seed metering device. This process directly drops to the seed bed without the guide components. Generally, the seeds in this method have a horizontal dividing speed relative to the backward device after the action of mechanical construction in the dropping process. The technology is now the most commonly used for potato planting model forms, including pneumatic-type device, moving-belt-type device, needle-type device, and the pickup-finger device using this method. By mechanical action, a kind of potato is made when leaving the metering device, which has a seeder with an initial velocity of movement in the opposite direction. Low-position direct seeding is low in height and does not require the installation of additional seed guide devices, but it is only suitable for low-speed seeding operations. It cannot meet the development trend of today's high-speed agriculture.

### 2.4.2. Seed Tube Seeding

The seed tube is adopted for maintaining seeds fallen from the seed metering device along its barrier to retain the uniformity of the seed quality. The seed tube can accelerate the speed in the opposite direction of movement of planter so that the seed can obtain the horizontal fractional rate in the opposite direction of the forward speed of the planter and then approach zero speed seeding [46]. Its structure and installation position on the seed tuber separation unit.

### 2.4.3. Air Blowing Seeding

The existing zero-speed seeding technology has many constraints, such as being unsuitable for high-speed operation and low seeding precision. In order to address these issues, a unique method of zero-speed seeding was presented by Lü et al. [65]. Positive pressure was used to make the seed accelerate in the opposite direction of seeding so that its velocity was zero relative to the seed bed. The precision of seeding was improved as well as the efficiency of seeding in this route.

## 3. Machine of Mechanized Planting

### 3.1. Soil Preparation Machine

The flipped plow is used for potato rotary tillage, which can complete the work of soil loosening, weeding, and so on. Both subsoiler and combined machines for subsoiling and soil preparation can realize all-directional sub-soiling.

Soil cultivators are a vital part of assuring potato planting. Effective soil cultivation maintains the quality of the soil and improves the retention of organic substances such as air and water. Some typical machines are shown in Table 4.

**Table 4.** Some typical soil preparation machines.

| Model | Structure | Characteristic |
|---|---|---|
| Shandong Transce Agricultural Machinery 1SL-6A subsoiler [66] | 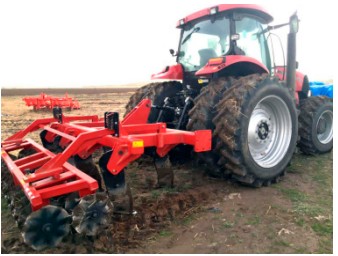 | Matched type, mounted; six seeding plough; minimum required power, 135 hp; pure weight, 1500 kg; The depth of the sub-soiling, 400–500 mm. The subsoiling shovel adopts a particular arc inverted ladder-type design. The operation does not disturb the soil and does not turn over the soil to achieve all-round Subsoiling. |
| Shandong Transce Agricultural Machinery 1LF-550 flipped plow [67] | 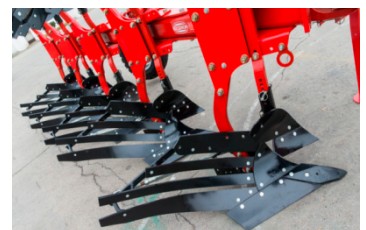 | Matched-type, mounted; six seeding plows; minimum required power, 65 hp; pure weight, 1300 kg; working width, 132/152/176/200 cm; the depth of tillage, 350–400 mm. The hydraulic piston rod drives the positive and negative plow on the plow frame to make vertical turnover movement and replace it alternately to the working position. |
| BOMET U473 Cultivators Dorado [68] | 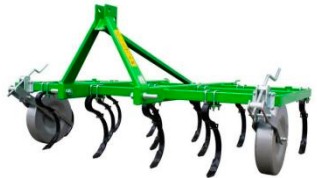 | Matched-type, mounted; three rows; minimum required power, 38 hp; working depth, 13 cm; pure weight, 180 kg; working width, 210 cm; the depth of tillage, 130 mm. The essential equipment of the machine is two supporting wheels that set the machine's working depth. |
| SPEDO VB4F/75 Bed Former [69] | 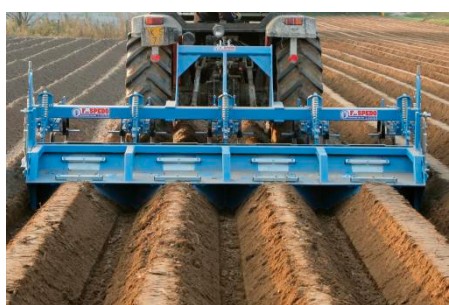 | Matched-type, mounted; four rows; minimum required power, 70 hp; pure weight, 720 kg; distance between rows, 80 cm; working width, 340 cm; working length, 200 cm. The flex springs, positioned on the front part of the bed former, move the land allowing the furrowing plow discs to form regular and uniform beds. |

**Table 4.** *Cont.*

| Model | Structure | Characteristic |
|---|---|---|
| GRIMME GF400 [70] | 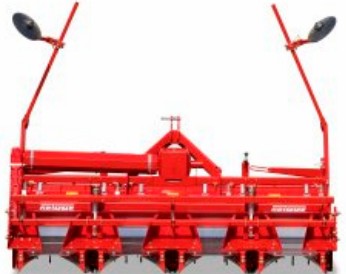 | Matched-type, mounted; four rows; minimum required power, 134 hp; pure weight, 1800 kg; row width, 75–90 cm; working width, 3–3.6 m. The machine can also be used in combination with a potato planter, thus enabling soil cultivation and planting in a single pass. For soil cultivation and seeding of fine seeds in a single pass, the machine can be equipped with hydraulically driven ridge pressure rollers and a lifting frame for seeders. |

*3.2. Potato Planter*

3.2.1. Cup-Belt (Chain) Type Potato Planter

Cup-belt (chain)-type potato planters rely on a simple structure, have a reliable performance, are easy to operate in all kinds of seeding machines, and have been widely used. With a combination of a simple structure, wide application, and innovative technology, many typical planters have been developed. Some common potato planters are shown in the Table 5.

**Table 5.** Introduction of Cup-belt (chain) type planters.

| Country | Model | Structure | Characteristic |
|---|---|---|---|
| Germany | Grimme GL420 [71] | 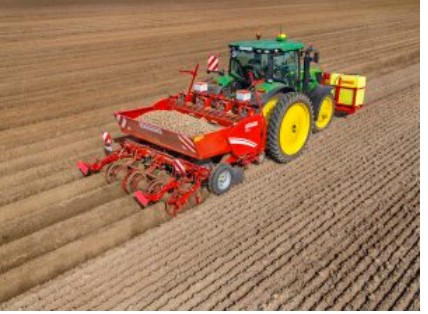 | Matched-type, trailed; four seeding rows; minimum required power, 120 hp; distance between rows, 75–90 cm; seed box capacity, 2 t. Equipped with a hydraulic control system and electronic monitoring system, combination with a cultivator is possible for optimum soil preparation. |
| USA | Double L9560 [12] | 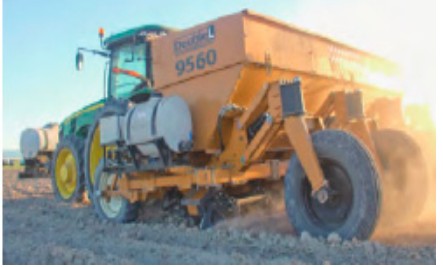 | Matched-type, trailed; six seeding rows; matched power, 168 hp; distance between rows, 71.12–101.6 cm; seed box capacity, 6.6 t. Optical seed sensors are used to monitor seed picking. The planter is equipped with optical sensors and GPS to ensure seeding accuracy. |
| Holland | Dewulf CP 42 [72] | 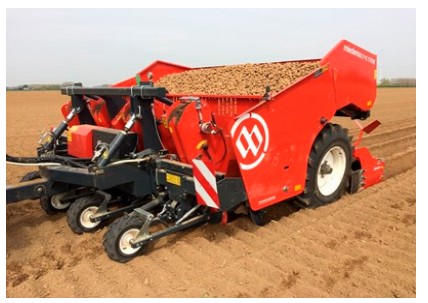 | Matched-type, trailed; four seeding rows; distance between rows, 75–90 cm; seed hopper capacity, 3.6 t. It is mechanically or hydraulically driven and equipped with a human–computer interaction system and electronic monitoring system. |

**Table 5.** *Cont.*

| Country | Model | Structure | Characteristic |
|---------|-------|-----------|----------------|
| China | Menoble 1240 [73] |  | Matched-type, mounted; four seeding rows; minimum required power, 100 hp; distance between rows, 80/90 cm; seed box capacity, 1.2 t. Seeding unit equipped with electronic vibration mechanism equipped with hydraulic control rowers. |
| Italy | Spedo SPA/A [74] |  | Matched-type, mounted; two seeding rows; distance between rows, 70–90 cm; required power, 50 hp; two seeding units; weight, 480 kg; seed box capacity, 500 kg; seeding velocity, 0.4–0.6 ha/h. |
| Poland | Bomet Sp. z o.o. Sp. K. [75] |  | Matched-type, mounted; matched power, 20 hp; single one seeding row; distance between rows, 290/320/350 mm; planting depth, 100–150 mm; pure weight, 130 kg. Changing the distance of seed potatoes in a row can be achieved by changing the diameter of the wheels. |
| India | SWAN AGRO NSE PPR-2 [76] |  | Matched-type, mounted; two seeding rows; distance between rows, 60–66 cm; required power, 40–60 hp; seed hopper capacity, 0.24 t; planting depth, 130–150 mm; working efficiency, 0.51 ha/h. |

3.2.2. Pneumatic-Type Potato Planter

At present, there are few commercial potato planters using pneumatic seed potato separation and whole row picking technology. They include the PLMS series pneumatic drill produced by the French ERME company and the Lockwood600 series produced by the American Crary company (including 604, 606, 608 three models) air suction potato planter. Take the Lockwood 606p planter as an example. The planter can finalize ditching, sowing, fertilization, drip irrigation, soil covering, repression, and finalize operations simultaneously. Some typical potato planters are shown in the Table 6.

**Table 6.** Introduction of belt-type potato planters.

| Country | Model | Structure | Characteristic |
|---------|-------|-----------|----------------|
| USA | Crary Lockwood 606 [77] | | Matched-type, mounted; six seeding rows; minimum required power, 20 hp; distance between rows, 80–100 cm; weight of the machine, 5 t; Seed hopper capacity, 5.44 t; maximum planting speed, 4.5 mph. The planter is equipped with hydraulic drive device and GPS navigation system, which can ensure high working speed and sowing quality. |
| Germany | Grimme Pneumatic precision mini-tuber planter | | Matched-type, mounted; double seeding rows; minimum required power, 120 hp; distance between rows, 80–100 cm, the machine itself is 1.5 T. maximum working speed, 8.2 km/h. It uses a vertical disc potato row, has strict requirements on the uniformity of the appearance and size of potatoes, and has a high rate of heavily missed seeding. |
| French | ERME PLMS planter [78] | | Matched-type, mounted; four seeding rows; minimum distance between rows, 35 cm. Drive wheels adjustable in height. |

### 3.2.3. Moving-Belt-Type Potato Planter

At present, the representative seeders using this technology include Grimme GB series belt seeders and Miedema Structural series produced by Dewulf. Some typical potato planters are shown in the Table 7.

**Table 7.** Introduction of moving-belt-type Planter.

| Country | Model | Structure | Characteristic |
|---------|-------|-----------|----------------|
| Germany | Grimme GB 230 [58] | | Matched-type, mounted; minimum required power, 120 hp; distance between rows, 70–90.4 cm; maximum working speed, 25 km/h; seed hopper capacity, 3 t. Seeding parameters and seeding quantity can be adjusted through the operator terminal. |

**Table 7.** *Cont.*

| Country | Model | Structure | Characteristic |
|---------|-------|-----------|----------------|
| Holland | Dewulf Miedema Structural 30 [79] | 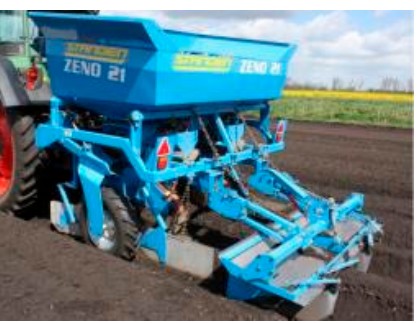 | Matched-type, trailed; three seeding rows; minimum required power, 80 hp; distance between rows, 4–100 cm; seed box capacity, 3.5 t; maximum working speed, 11 km/h. The inclination of the seed box is adjusted by a hydraulic drive. |
| England | Standen Engineering ZENO 21 [80] | 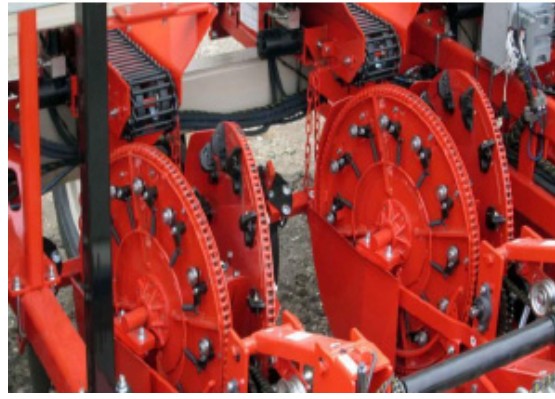 | Matched-type, mounted; two seeding rows; matched power, 102 hp; distance between rows, 76–102 cm. Equipped with a hydraulic control system to control depth. The uniformity of sowing is ensured by monitoring the speed with sensors. |

3.2.4. Needle-Stabbed Type Totato Planter

The research on the technology and equipment of needle-stabbed is mainly concentrated in the USA. A number of technological invention patents centering on the research and development of needle-stabbed were applied in the last century. Because this type of seeding device can easily cause bacterial infection, the application of this technology is less at present. The typical model is the Lockwood 6200 series planter [81] (Figure 15) and Harrison pick planter [82] (Figure 16).

**Figure 15.** Lockwood 6200 Needle-type potato planter. Reprinted with permission from ref. [48] Copyright 2018 Lockwood Industries.

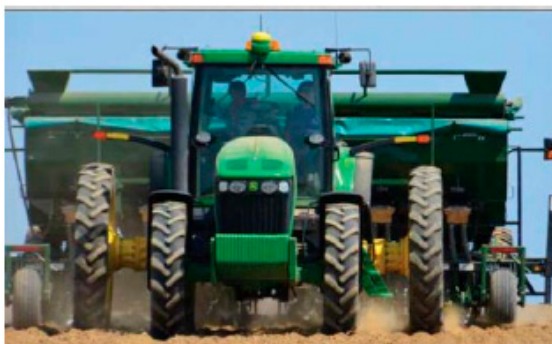

**Figure 16.** Harriston Pick Planter. Reprinted with permission from ref. [48] Copyright 2019 Harriston Industries.

### 3.2.5. Pickup-Finger-Type Potato Planter

This type of planter has poor performance stability and operation quality. The curve of the finger clip guide rail of this structure is fixed, resulting in the fixed opening stroke of the finger clip. However, the overall dimensions of potatoes are different, so there is a high rate of reseeding and missed seeding, and it is impossible to operate at high speed. At present, this principle has been applied in the seed metering device with small particle size and regular shape for corn and soybean, while in the potato planter, only a few manufacturers, such as Lockwood and Harriston, have provided models that can be applied practically. The Lockwood 506 series potato planter is presented in Figure 17, the Harriston Clamp planter [83] is presented in Figure 18.

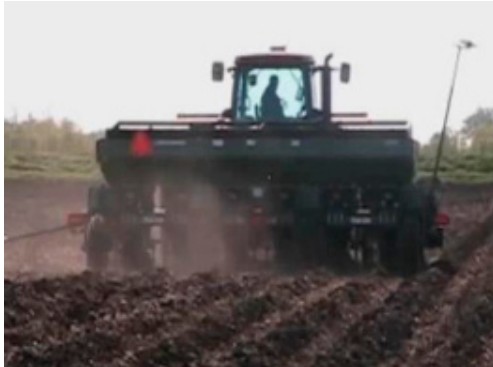

**Figure 17.** Lockwood 506 series potato planter. Reprinted with permission from ref. [48] Copyright 2018 Lockwood Industries.

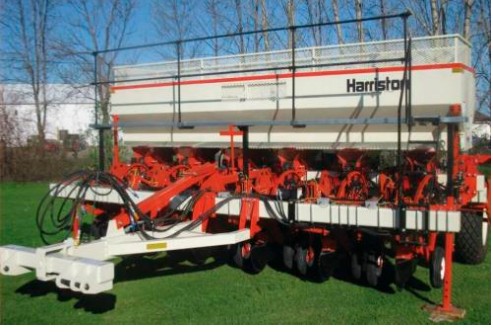

**Figure 18.** Harriston Clamp planter. Reprinted with permission from ref. [83] Copyright 2019 Harriston Industries.

3.2.6. Vibration Separation-Type Potato Planter

Due to the limitation of manual capacity, the working speed of the planter using the vibration separation principle will not exceed 2 km/h. Figure 19 shows the Japanese JAGIRL vibrating separation row artificial cleaning (supplementary) seed potato planter, which is suitable for sowing in small plots.

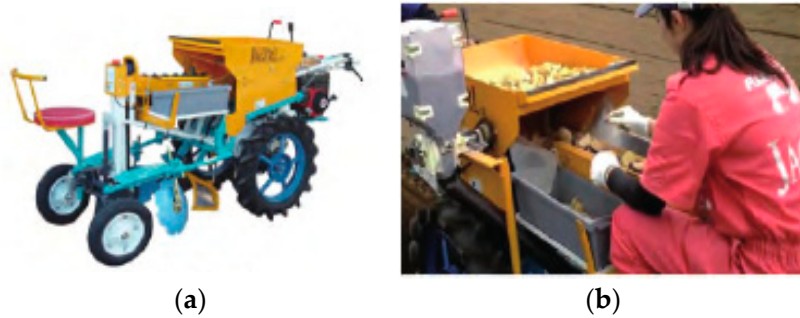

(**a**)　　　　　　　　　　　　　　　　(**b**)

**Figure 19.** JAGIRL man-aided potato planter (**a**) The structure of planter. Reprinted with permission from ref. [13] Copyright 2019 JAGIRL Industries (**b**) Manual assisted operation. Reprinted with permission from ref. [13] Copyright 2019 JAGIRL Industries.

At present, there are many types of potato planters for different regions; but the mechanization degree of small potatoes in hilly and mountainous areas is insufficient; which will be an important direction for future research and development. The types of planters in Asia are mainly traditional planters; while the development of planters in the United States and other regions is mainly focused on precision and high speed; and the application of many types of planters is also mature

**4. Research on Intelligent Seeding Technology**

The concept of the intelligent system of the planter was proposed as early as the 1940s, with research and development on intelligent control gradually conducted [84]. However, it was only until the recent four decades that intelligent seeding technology fully utilized the advances in sensor, artificial intelligence, and electrical driving technology [85,86]. Intelligent seeding technology requires the planter to meet the three conditions of precise seeding rate, spacing and seeding depth to sow the seed potato to the desired depth accurately.

*4.1. Seeding Monitoring System and Seeding Compensation System*

Miss seeding will reduce potato yields and soil efficiency, and the rate of missed seeding is reduced by improving the structure of the planter. However, the rate of missed seeding of the improved potato planter is still beyond 5% [87]. The simplest way to reduce this loss was manually assisted reseeding, but it required excellent labor intensity and reduced efficiency. Therefore, the current standard method is to use the monitoring system to monitor the miss seeding and add the compensation device [88].

The miss seeding rate of large potato planters is low, and the existing achievements mainly focus on essential information collection and alarm indication. For example, Grimme GB series planter, Dewulf CP series planter, and Double L series planter are equipped with photoelectric sensors [12] (Figure 20a) to monitor the process of seeding. The detection system generally uses the photoelectric sensor and distance sensor to monitor the seeding condition of the seeding unit. When the seeding is abnormal, the Single-Chip Microcomputer controls the alarm system to prompt the driver to eliminate the fault. When miss seeding is detected, the Single-Chip Microcomputer drives the compensation system to reduce the miss seeding and ensure the seeding quality. Advanced GPS technology has also been introduced into the Double L monitoring system.

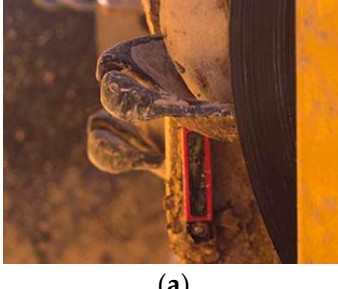 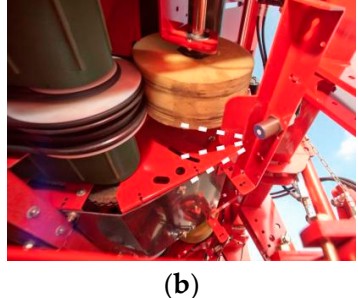 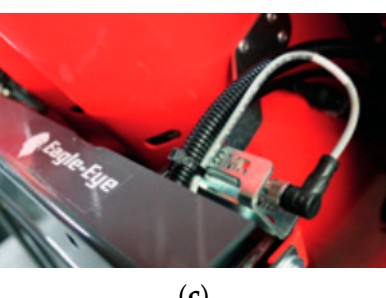

(**a**) (**b**) (**c**)

**Figure 20.** Seeding monitoring sensor: (**a**) Double L photo electric sensor. Reprinted with permission from ref. [12]. Copyright 2021 Double L Industries; (**b**) GB430 ultrasonic sensor Reprinted with permission from ref. [89] Copyright 2022 Grimme Industries; (**c**) Dewulf CP series Eagle-Eye. Reprinted with permission from ref. [72]. Copyright 2021 Dewulf Industries.

For the Grimme GB430 planter [89], the seeding distance is monitored by the ultrasonic sensor (Figure 20b), which counts the seeds as they fall into the seed bed and detect whether there is deviation at the same time. When a deviation occurs, it is fed back to the driver through the monitoring system and even adjusted [81].

For Dewulf CP series planter [72], the 'Eagle-Eye' (Figure 20c) registers each tuber. Two sensors for the issuing of the signal to the control terminal located in the cabin. The user can set a limit beforehand for the percentage of misses. When this threshold is exceeded, the system will issue both a visual as well as an audible signal. As a result, the user can check and modify the planting adjustments.

In addition to seeding monitoring, sensors are also used in other aspects of potato planters, such as the Dewulf Miedema series planter [79], a tipping automat sensor (Figure 21) that detects the number of potatoes in the seed box. Then the machine can ensure that the supply belts are supplied with a sufficient quantity of seed potatoes.

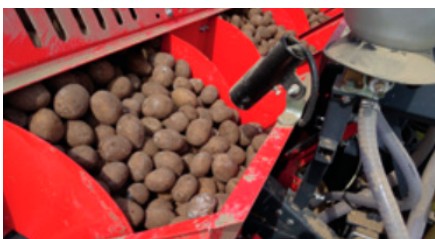

**Figure 21.** Tipping automat sensor. Reprinted with permission from ref. [80]. Copyright 2021 Dewulf Industries.

At present, the small and medium-sized potato seeder with a chain spoon seed metering device has a high rate of missed seeding, so the current related research focuses on intelligent and automatic monitoring system research. In order to ensure the anti-dust interference detection performance of the sensor, the detection sensor was installed in the upper section of the tube, and the seed cup was parallel to the horizontal plane. The transmitter and receiving end of the sensor should be installed on the same horizontal plane (as shown in Figure 22), and the center should be aligned. The sensor was installed on a row of tubes, and the installation spacing was determined according to the distance between cups.

Zhang et al. [91] proposed an electromechanical potato planter automatic compensation system composed of an infrared photoelectric sensor, Single-Chip Microcomputer, and stepper motor. Still, its missed seeding detection scheme was primitive and had low reliability. On this basis, a potato missed seeding compensation system based on an infrared miss-seeding detection and the AT90S2313 SCM was organized by Liu et al. [92]. Cao et al. [90] completed the hardware design of a miss-seeding early warning system,

including DC regulated power supply design, sensor module, MCU control module, alarm module, and display module. System software used C language to write the system monitoring program. A technique of monitoring potato missed seeding by infrared radiation was proposed by Sun and Wang et al. [93,94]. They also proposed a new architecture of two-point monitoring information statistics and ranking decisions, which makes the system response more advanced and overcomes the constraints of the first-generation detection technology on sensor installation positions. However, the system is still vulnerable to the threat of field dust, vibration, and other external factors, and its dependability still needs to be enhanced. To solve this problem, a new approach was proposed to construct a space capacitance sensor for the evaluation of seed-metering states and mass acquisition of seed potatoes by Zhu et al. [95]. Wang.G. et al. [96] proposed a kind of integrated seeding and compensating potato planter based on one-way clutch (Figure 23), a missed seeding and compensation system using an infrared radiation type missed seeding detection system to realize a one-way clutch and motor matching.

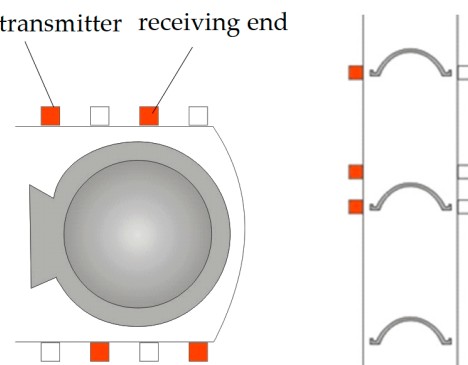

**Figure 22.** Monitoring location. Reprinted with permission from ref. [90]. Copyright 2013 Northwest A&F University.

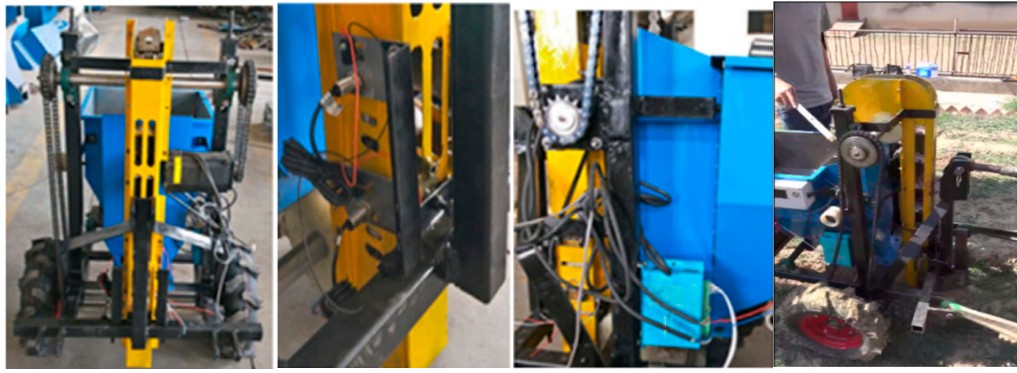

**Figure 23.** Integrated seeding and compensating potato planter based on one-way clutch. Reprinted with permission from ref. [96]. Copyright 2019 Gansu Agricultural University.

Pneumatic type device also has the situation of missed seeding, the intelligent seed supply system of the air-suction potato planter (Figure 24) was designed by Lü et al. [18]. The controller collects information through the seed box weighing sensor and calculates and outputs the results to speed the stepping motor to control the seeding speed.

At present, the monitor and compensation control technology are mainly reflected in the material level detection device. The ultrasonic detection technology, infrared sensing technology, or image recognition technology is used to accurately and stably identify the number change of seed potato in the seed box, which can timely and stably drive the seed supply device for seed supply. Secondly, the structure principle of seed supply device is different, but its development trend is to provide stable seed precision and reduce the damage of seed potato.

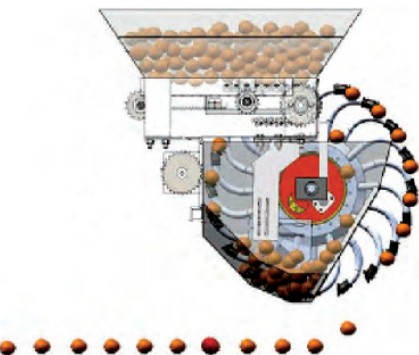

**Figure 24.** The intelligent seed supply system of the air-suction potato planter. Reprinted with permission from ref. [18] Copyright 2022 Northeast Agricultural University.

*4.2. Seeding Depth Control*

In the process of potato planting, due to the changes in the terrain in the field, the different properties of the soil, and the remaining stems and stubble on the ground, the sowing depth is difficult to be unified, resulting in the potato yield being affected.

The automatic hydraulic depth control is performed conveniently from the operator terminal in a large potato planter. Such as Grimme GB430 [89], trailed furrow openers are mechanically connected and guided in a parallelogram, as shown in Figure 25a. Moreover, change the number of feeler wheels used to pull furrow openers to suit different soil conditions. For lighter soils, the pulled furrow openers are individually guided in height by a feeler wheel, as shown in Figure 25b. For heavy soils, the pairwise mechanically connected, pulled furrow openers are guided in the depth by two large feeler wheels, as shown in Figure 25c.

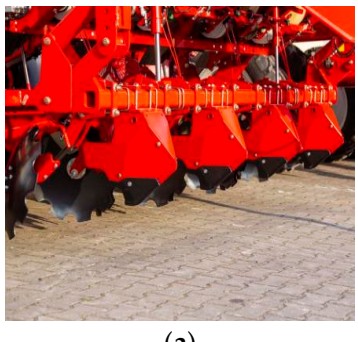
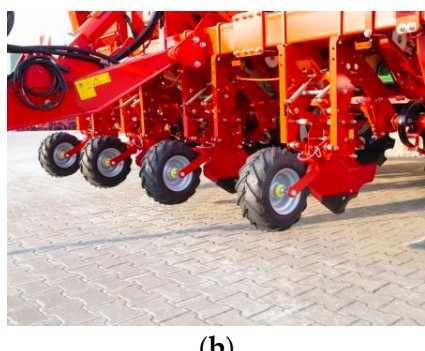
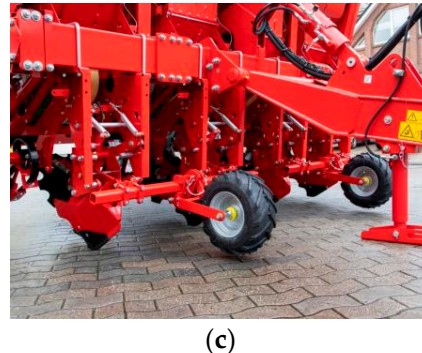

(**a**)          (**b**)          (**c**)

**Figure 25.** Grimme GB 430 potato planter. (**a**) Trailed furrow openers of GB430. Reprinted with permission from ref. [89] Copyright 2022 Grimme Industries; (**b**) Structure for lighter soils. Reprinted with permission from ref. [89] Copyright 2022 Grimme Industries; (**c**) Structure for heavy soils. Reprinted with permission from ref. [89] Copyright 2022 Grimme Industries.

Dewulf Miedema CP 42 planters [72] (Figure 26) are equipped with an intelligent floating control system, which can monitor the depth of the seed trench timely and automatically adjusts the soil tillage depth through sensors during the seeding process so as to automatically maintain a certain tillage depth while ensuring sufficient seeding precision.

Grimme company mainly uses feeler wheels to pull the furrow openers to ensure the constant seeding depth. In the GL-34T potato planter [97] (Figure 27) suitable for large slopes, the lead screw with an adjustable depth limiting wheel is used to ensure the uniformity of seeding depth.

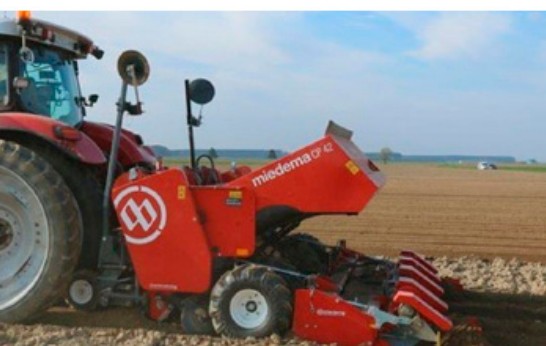

**Figure 26.** Dewulf Miedema CP 42 potato planter. Reprinted with permission from ref. [72]. Copyright 2021 Dewulf Industries.

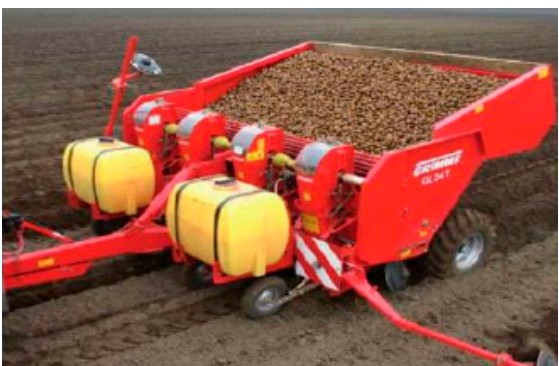

**Figure 27.** Grimme GL-34T potato planter. Reprinted with permission from ref. [97]. Copyright 2018 Grimme Industries.

Zou et al. [98] designed a potato profiling ridging sowing device for potato seeding in dry land ridge cultivation, which can ensure a constant soil seeding depth through the soil covering profiling mechanism and limit sowing of the seedbed. Hu et al. [99] used a parallelogram ditcher to ensure the consistency of seeding depth in a 2CMW-4B micro potato planter, as shown in Figure 28.

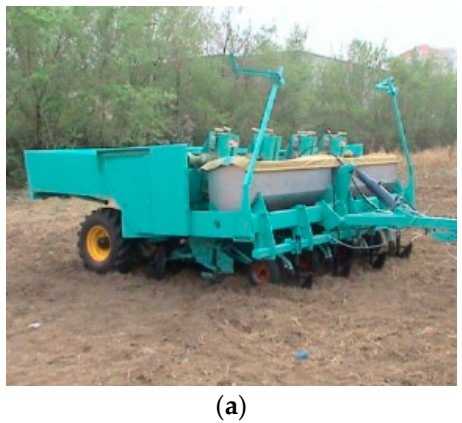
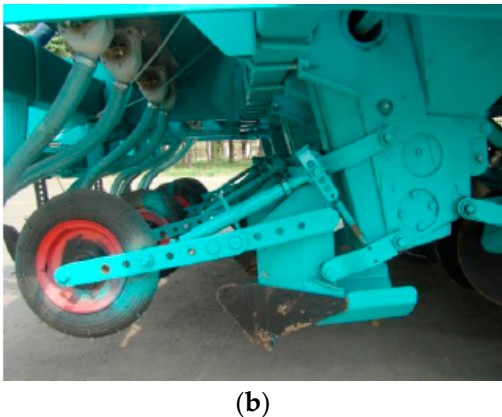

(**a**)　　　　　　　　　　　　　　　　　　　　　　　　(**b**)

**Figure 28.** Two views of the 2CMW-4B planter and the depth control device. (**a**) Operation process of the device; (**b**) Structure of ditching and seeding device. Reprinted with permission from ref. [99]. Copyright 2019 Shandong University of Technology.

## 4.3. Electrical Driving Technology and GPS

For large potato planters, automatic control or manual adjustment of seeding parameters by the driver is usually adopted, such as, Field-Ready Controller (Figure 29) was used

in the Double L9560 planter [12], the controller receives GPS, radar, and ground speed data from the tractor. The controller is usually used to control the hydraulic system to drive the seeding device. These parameters are mainly used to adjust the seeding spacing and fertilization amount, with control the number of seeds entering the seed metering device by monitoring and calculating the sowing conditions. The electrical driving system is performed conveniently from the HMI operator terminal, as shown in Figure 30.

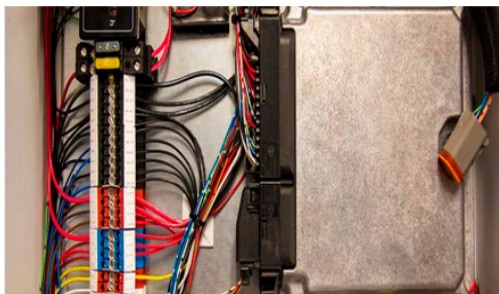

**Figure 29.** Field-Ready Controller. Reprinted with permission from ref. [12]. Copyright 2021 Double L Industries.

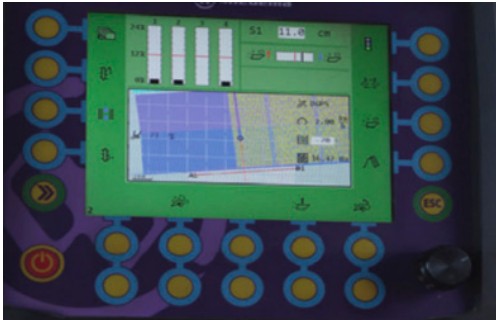

**Figure 30.** HMI operator terminal. Reprinted with permission from ref. [12]. Copyright 2021 Double L Industries.

For the Grimme GB430 potato planter [89], all planting components are jointly operated by a single hydraulic motor (Figure 31). The hydraulic drive can confirm the direct stepless adjustment of the seeding spacing to adapt to the transformations in the roadway consistency. Each belt planting element is driven by its hydraulic motor.

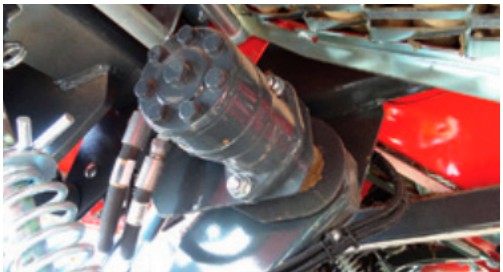

**Figure 31.** Hydraulic drive motor. Reprinted with permission from ref. [90] Copyright 2022 Grimme Industries.

Seed metering devices of potato planters are usually driven by a ground wheel and chain and sprocket system. This transmission scheme is easily affected by the slipping of the ground wheel and the vibration of the chain, so the uniformity of planting spacing cannot be guaranteed. Experiments have proved that the chain variation range of the slip rate of the ground wheel is as high as 20% [94].

In order to improve the seeding accuracy and ensure the seeding quality, some scholars proposed to use the motor to replace the ground wheel drive system. Wang et al. [94] used the hydraulic motor to drive the fine-tuning screw rotation to adjust the amplitude to achieve the step-less adjustment of the planting spacing. During the seeding operation, the hydraulic motor is driven by the hydraulic device of the tractor so as to drive the seed potato conveyor belt and seed potato cup. The device is equipped with two sensors. When one of them detects the seed potato bowl and the other does not detect the seed potato, that is, miss-seeding occurs, the main control module sends an adjustment command to the stepping motor to rotate the corresponding angle so as to adjust the vibration amplitude.

In order to improve the seeding accuracy of the potato planter, GPS technology is combined with the control system, as shown in Figure 32. By making use of GPS, the planter can map out the parcel. The machine will subsequently control that planting is performed accurately, allowing the driver to focus entirely on the planting process. For the farmer who places the very highest requirements on ease of use, efficiency, and precision, Dewulf has developed the GPS Planting-Comfort and GPS Planting-Control options. This easy-to-operate system automates many tasks.

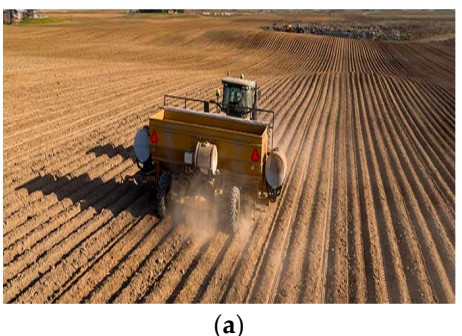 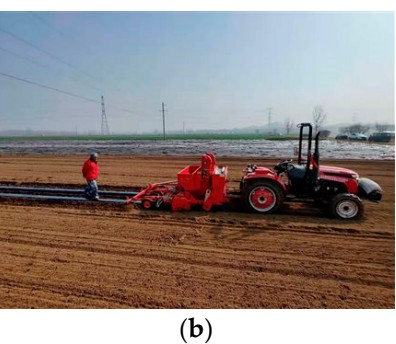

(**a**)          (**b**)

**Figure 32.** Automatic Potato planter with GPS system. (**a**) Double L planter works under GPS navigation. Reprinted with permission from ref. [12]. Copyright 2021 Double L Industries. (**b**) Navigation test of planter.

## 5. Conclusions and Recommendations

Currently, potato is planted in a wide range of regions all over the world. The difference in environmental characteristics in each region leads to different planting modes and different agronomic requirements. Therefore, the development degree of planting mechanization is also different. In terms of the critical technologies of mechanized planting, the research on the characteristics of seed potatoes has made some achievements. Whereas various planters are marketed around the world, research is still ongoing. After years of development, some developed countries have used a large number of advanced technologies such as automatic control, hydraulic system for seed supply, seeding electronic monitoring, etc., with a high degree of intelligence and precision. However, the level of mechanization of potato planting in hilly mountain areas is relatively backward, and the yield of potatoes is not high. Accordingly, the following suggestions are put forward.

The breeding of seed potatoes plays a very important role in increasing potato yield. In order to improve potato yield, it is necessary to introduce new potato varieties with high yield, high resistance, and special purpose in some potato production areas, select high-quality varieties suitable for planting in various planting areas and promote characteristic potato varieties according to the conditions of potato planting areas. Secondly, it is necessary to increase the use of virus-free mini-tuber.

Accelerate the research and development of small and medium-sized machinery that can realize the whole process mechanization of potato, solve the sectional operation research, and support small machinery selection of mechanical seeding and mechanical harvesting, and accelerate the research and development process of miniature and medium-sized agricultural machinery and agronomic integration technology of potato. For hilly

and mountainous areas, strengthen the mechanization research and machinery selection under the conditions appropriate for viscous soil, and focus on the power chassis suitable for hilly and mountainous areas and small-scale tillage and land preparation machinery in Hilly and mountainous areas.

For the problems of poor seeding quality and stability of the current seed metering device, the mechanism of each type of seed metering device is further studied. At the same time, advanced technologies, new materials, and manufacturing procedure are continuously integrated into the manufacturing process of the seed metering device and the whole machine to improve the stability and reliability of the potato planter performance and develop new potato seed metering devices and seeding device.

In the future, the development of potato mechanized planting technology and equipment will focus on the precision, high-speed, intelligent large-scale potato planting equipment, and the synchronous research and development of economic, light, and simple potato mechanized planting technology and equipment in some particular areas will be the main development direction. Improving the automation and intelligence level of potato planters will be the focus of future research. In future studies, digital technology will become an essential part of improving work quality and efficiency.

**Author Contributions:** Conceptualization, B.Z., Y.L., C.Z. and Q.N.; validation, C.L. and L.C.; investigation, Y.L.; resources, S.X. and Q.N.; data curation, B.Z. and C.Z.; writing—original draft preparation, B.Z. All authors have read and agreed to the published version of the manuscript.

**Funding:** This work was supported by the Graduate Scientific Research Innovation Project of Chongqing, China (CYS22216), the Fundamental Research Funds for central Universities (SWU120004), Natural Science Foundation of Chongqing, China (cstc2021jcyj-msxmX1178) and the Science and Technology Plan Project of Sichuan, China (2021YFQ0070). The authors take full responsibility for the content of this paper.

**Data Availability Statement:** The study did not report any data.

**Conflicts of Interest:** The authors declare that they have no known competing financial interests or personal relationships that could have appeared to influence the work reported in this paper.

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
