# Peer review of "Potato Planter and Planting Technology: A Review of Recent Developments"

_agriculture, doi:10.3390/agriculture12101600_

Round 1

Reviewer 1 Report

The aim of this article was to review the recent developments on the potato planter and planting technology. I suggest minor revision for this manuscript based on the specific comments below.

1.      ‘Originated’, ‘The United States’, ‘Mechanized’, ‘Potato’, ‘Should’,… Please check the capital letter in these words.

2.      ‘With the continuous decline of the sown area and production in Europe and the United States and the continuous development and increase in Asia and other regions’… There are too many ‘and’ in this sentence, please rewrite it.

3.      Does the word ‘continents’ correct in table 2?

4.      ‘This paper is organized as follows…’ Please rewrite this paragraph.

5.      ‘Figure 2 presents three kinds’ …three kinds of seed tubers.

6.      ‘the shape index () of seed tubers is obtained through the formula’ In this sentence, which character was used to express the shape index? How to calculate the shape index? Which formula was used to calculate the shape index?

7.      Can the authors provide some detail information on the physical characteristics of seed potatoes?

8.      Since the sliced potato was the most commonly used potato in China, how to establish the model of sliced potato using EDEM?

9.      Give more information about Fig. 9 (a) and (b), Fig. 10 (a) and (b), Fig 14 (a) and (b), and Fig 20 (a) and (b).

10.  Delete ‘and small plots’ in line 408.

11.  The picture title of Fig. 24 and Fig. 25 was not correct.

12.  Please added appropriate references for the MS from line 430-467, line 505-528, line 536-553, and line 569-576.

13.  All the picture titles were too simple. All the pictures in the MS should given the references.

Author Response

Thank you for reviewing our paper. We appreciate your insightful comments. Those comments are all valuable and very helpful for revising and improving our paper. We have studied comments carefully and have made correction which we hope meet with approval. Papers are uploaded in subscription mode, with major revisions in red.The reply was placed in the attachment.

Reviewer 2 Report

This review presents the research progress and application status of potato planters and planting technology worldwide. It classifies the planting technology into four types, research of material characteristics for potatoes, soil cultivation, seed potato separation, and zero-speed seeding. The review provides enough details about potato mechanization. The manuscript in its present form is suitable for publication in Agriculture journal after some suggestions and comments that are given in the attached file. 

Author Response

Thank you for reviewing our paper. We appreciate your insightful comments. Those comments are all valuable and very helpful for revising and improving our paper. We have studied comments carefully and have made correction which we hope meet with approval. Papers are uploaded in subscription mode, with major revisions in red. The reply was placed in the attachment.

Reviewer 3 Report

The submitted manuscript (Review) has been prepared with due care. The authors have carefully reviewed the extensive literature and made a logical argument.

I note the adaptation of the manuscript to the editorial requirements of the journal. 

Author Response

(The authors gave the same response as above.)
